# SMOOTHIE: Label Free Language Model Routing

Neel Guha [*]    Mayee F. Chen[*]    Trevor Chow    Ishan S. Khare    Christopher Ré

Stanford University, Department of Computer Science

{nguha, mfchen, tmychow, iskhare, chrismre}@stanford.edu

## Abstract

Large language models (LLMs) are increasingly used in applications where LLM inputs may span many different tasks. Recent work has found that the choice of LLM is consequential, and different LLMs may be good for different input samples. Prior approaches have thus explored how engineers might select an LLM to use for each sample (i.e. *routing*). While existing routing methods mostly require training auxiliary models on human-annotated data, our work explores whether it is possible to perform *unsupervised* routing. We propose SMOOTHIE, a weak supervision-inspired routing approach that requires no labeled data. Given a set of outputs from different LLMs, SMOOTHIE constructs a latent variable graphical model over embedding representations of observable LLM outputs and unknown "true" outputs. Using this graphical model, we estimate sample-dependent quality scores for each LLM, and route each sample to the LLM with the highest corresponding score. We find that SMOOTHIE's LLM quality-scores correlate with ground-truth model quality (correctly identifying the optimal model on 9/14 tasks), and that SMOOTHIE outperforms baselines for routing by up to 10 points accuracy.

## 1 Introduction

Large language models (LLMs) are increasingly being deployed in *multi-capability* regimes where data inputs may span a diverse range of tasks, each of which requires different capabilities [8]. For instance, an LLM-powered chatbot may be asked to write code, answer questions about different domains, summarize documents, perform extraction, and more [3, 8, 14, 30]. One challenge is that while engineers often have access to numerous pre-trained LLMs (i.e., through Huggingface or various APIs), they do not know which LLM is optimal for each possible user input [86]. Because the quality of generations can significantly vary across LLMs, choosing the right LLM for each input sample is important for ensuring high task performance [41].

Recent work has explored various ways to utilize ensembles of pretrained LLMs in multi-capability settings, by (1) collecting a diverse pool of LLMs and (2) identifying which LLM to *route* each sample to [55, 86]. However, the majority of existing approaches require labeled data; engineers typically either (1) train an auxiliary model using labeled data to rank or predict which LLM each sample should be routed to [41, 79], or (2) directly use labeled data to determine which LLM is the best on average [86]. As a result, engineers designing routing protocols face the practical difficulty of constructing labeled datasets.

Given a candidate pool of LLMs and an unlabeled test dataset, this paper explores how to best select LLM outputs for each sample in an entirely unsupervised manner—without labeled data, or models trained on labeled data. To make progress in addressing this question, we face two technical challenges:

- **Unknown LLM quality**: The first challenge is estimating the quality of each LLM. Access to labeled data allows engineers to identify higher performing LLMs by measuring the accu-

---

[*]Equal contribution.

38th Conference on Neural Information Processing Systems (NeurIPS 2024).

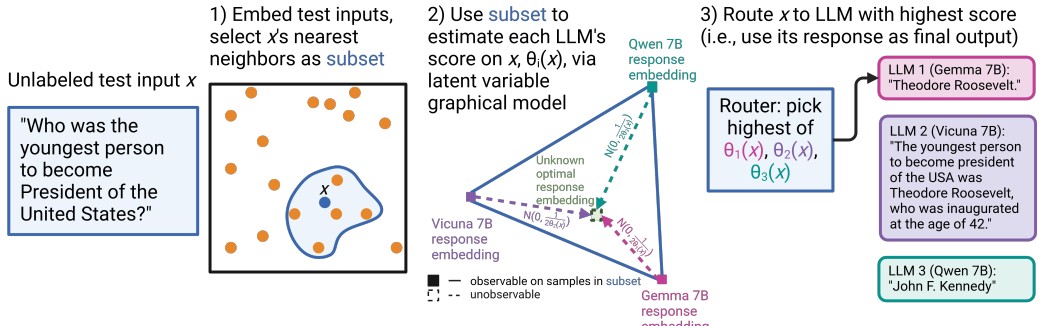

**Figure 1:** For a given input $x$, SMOOTHIE estimates the quality of every LLM ensemble's generation, and uses this quality weight to route $x$ to a single LLM.

racy/quality of LLM outputs. In this paper, we study the question of how to estimate quality *without* labeled validation data.

- **Sample-conditional generator performance**: The second challenge is determining how to select the best LLM for each individual test sample. LLM outputs can vary in quality over different samples, which could render population-level estimates of LLM quality misleading.

In this work, we propose SMOOTHIE, a method for routing samples to LLMs in a label-free manner (Figure 1). Below, we describe how SMOOTHIE addresses the two challenges described above.

- **Quality estimation**: Using the LLM outputs for each test sample as "voters," SMOOTHIE estimates the quality of each generator using methods from Weak Supervision (WS). Concretely, SMOOTHIE constructs a latent variable graphical model over observable LLM outputs and an unknown true output. By modeling the embedding vector difference between each LLM output and the true output as a multivariate Gaussian, we can derive a closed-form estimator adapted from [85] for learning LLM quality scores efficiently.

- **Conditioning**: We condition these quality estimates to be particular to a given test sample by only using the nearest neighbors of a test sample in the training data as inputs to the estimator (i.e., kernel smoothing). We then route each test sample to the LLM with the highest quality score estimate on that sample. We call the version of SMOOTHIE that produces quality estimates using all available test data SMOOTHIE-GLOBAL, and we call the version that uses a sample's nearest neighbors SMOOTHIE-LOCAL.

We empirically evaluate SMOOTHIE in three stages.

- **LLM selection:** First, we assess SMOOTHIE-GLOBAL's ability to identify—from an ensemble of mixed quality LLMs—the optimal LLM for a given task overall. On traditional generation tasks such as summarization, reading comprehension, and data-to-text generation, we find that SMOOTHIE-GLOBAL's learned LLM quality-weights correlate with actual LLM performance ($\rho = 0.72$)), and on the AlpacaEval benchmark, SMOOTHIE-GLOBAL identifies the best-performing instruction model 70% of the time [50]. The highest quality LLM identified by SMOOTHIE-GLOBAL—all computed without labeled data—can beat random-selection by up to 15 points win-rate on AlpacaEval, and by up to 8 points on SQuAD.

- **Routing:** Second, we study whether SMOOTHIE-LOCAL's sample-conditional scoring mechanism allows it to *route* samples in mixed-task datasets to higher-performing LLMs (i.e., the multi-capability regime). We find that SMOOTHIE-LOCAL can improve the quality of produced generations by up to 7 points accuracy over SMOOTHIE-GLOBAL, and that SMOOTHIE-LOCAL outperforms baseline unsupervised routing methods by up to 10 points accuracy and supervised routing methods by up to 5.0 points accuracy.

- **Prompt selection:** Finally, we assess whether SMOOTHIE's quality-estimation mechanism can be applied to select the optimal prompt template in a candidate pool while using a fixed LLM. We find that SMOOTHIE-GLOBAL can outperform other prompt selection approaches by up to 18 points, allowing a 410M parameter model to match the performance of 6.9B parameter model.

## 2   Related Work

We provide an abbreviated related work, with a full treatment in Appendix C.

**Routing** Routing has been classically utilized in Mixture-of-Experts models [25, 37, 42, 82], which involve jointly training a set of models as well as a router. Recently, routing mechanisms have been used at inference time to decide which pre-trained LLM to use for a given sample [79]. Some approaches involve training an auxiliary model using labeled training data to either score or rank the performance of each LLM on each test sample [38, 74]. Others do not involve training a model but instead use nearest neighbor methods, selecting the LLM that does the best on a test sample's labeled neighbors [48, 86]. In contrast, SMOOTHIE does not require any labels.

**Ensembling** Ensembling is another way of utilizing a pool of LLMs. Existing work has primarily focused on ensembling outputs for classification tasks [2, 68, 98]. Ensembling generative outputs typically requires training an auxiliary model [41], combining or switching among outputs when decoding [36, 83], or averaging in weight space [95].

**Prompt selection** In addition to selecting the best LLM for a sample, prior works have studied how to select the best prompt or in-context examples. While the simplest approach is to use a held-out labeled dataset [67], there are also retrieval-based approaches to selecting the best in-context examples [90], as well as approaches based on mutual information [89] and probability-based measures [103], although the latter two are limited to classification.

**Weak supervision** SMOOTHIE utilizes statistical techniques inspired by weak supervision, which programmatically generate labels for an unlabeled dataset by aggregating the predictions of several weak "voters" via a latent variable graphical model [71, 73]. Weak supervision has mostly been studied in classification settings [26, 72] but more recently has been extended to tasks such as learning rankings and manifolds [85, 94]. We derive our estimation procedure from the Gaussian model in [85], applying it to LLM embeddings and the routing setting.

## 3   Preliminaries

### 3.1   Problem setup

Let $\mathcal{V}$ be the token vocabulary space, and let $\bar{\mathcal{V}} = \mathcal{V} \times \cdots \times \mathcal{V}$ be the space of all vocabulary sequences. We consider a generative task with input text $x \in \mathcal{X} \subset \bar{\mathcal{V}}$ and reference output text $y \in \mathcal{Y} \subset \bar{\mathcal{V}}$. We have a candidate pool of $m$ LLMs, $G = \{g_1, \ldots, g_m\}$, where each $g_i \in \mathcal{G} : \mathcal{X} \to \mathcal{Y}$ produces a generative output sequence $g_i(x)$ for a given input text sequence $x$. We are given an unlabeled test dataset $\mathcal{D}_{\text{test}} = \{x_i\}_{i=1}^n$, where the ground-truth reference outputs are *unknown*.

Our goal is to route each sample $x \in \mathcal{D}_{\text{test}}$ to one of the LLMs in $G$. Specifically, we wish to construct a router $\text{route} : \mathcal{G}^m \times \mathcal{X} \to \mathcal{G}$ that selects the LLM that yields the highest quality generation on $x$ for each test sample $x$, without any labeled data.

### 3.2   Graphical model

We present a probabilistic graphical model (see Figure 1 (center)) that describes how the LLM outputs, $g_1(x), \ldots, g_m(x)$, are related to a true output $y$ in terms of each LLM's quality on a given input $x$, which we call $\theta_i(x)$, corresponding to each $g_i(x)$. Let $z_{g_0} : \bar{\mathcal{V}} \to \mathbb{R}^d$ map from a sequence of tokens to a $d$-dimensional embedding using a common model $g_0$ such as SentenceBERT [76]. Define $\lambda_i(x) := z_{g_0}([x, g_i(x)])$ to be the observable embedding of $x$ and the LLM output, and define $z^\star(x) := z_{g_0}([x, y])$ to be the latent ground-truth embedding of $x$ and reference output $y$. Similar to the approach in [85], we model the distribution over embedding vectors, $\Pr(z^\star(x), \lambda_1(x), \ldots, \lambda_m(x)|x)$ as

$$\Pr(z^\star(x), \lambda_1(x), \ldots, \lambda_m(x)|x) = \frac{1}{Z} \exp\left( \sum_{i=1}^m -\theta_i(x)\|\lambda_i(x) - z^\star(x)\|^2 \right) \tag{1}$$

where $Z$ is the log partition function and the $\theta_i(x)$s—the LLM quality scores—are canonical parameters of the graphical model. Intuitively, our model captures LLM quality by supposing that if $g_i$ is of high quality and $\theta_i(x)$ is very large, then it should be unlikely for the LLM output to be very

---

**Algorithm 1** ESTIMATE SCORES

---

1: **Input:** unlabeled test dataset $\mathcal{D}_{\text{test}}$, LLMs $G$, $n_0$ nearest neighbors parameter, $g_0$ embedding model with dimension $d$.
2: For all $x \in \mathcal{D}_{\text{test}}$ and $g_i \in G$, obtain the generator output $\vec{g}_i(x)$ and embed the input and generator output using model $g_0$ to get embedding $\lambda_i(x) := z_{g_0}([x, g_i(x)])$.
3: **for** $x \in \mathcal{D}_{\text{test}}, g_i \in G$ **do**
4:     **for** $j, k \neq i \in [m]$ **do**
5:         Compute $\hat{\delta}_{ij}(x) = \frac{1}{n_0} \sum_{x' \in \text{NN}_{n_0}(x)} \|\lambda_i(x') - \lambda_j(x')\|^2$, and similarly $\hat{\delta}_{ik}(x)$ and $\hat{\delta}_{jk}(x)$.
6:         Set $\hat{\theta}_i^{jk}(x) = d/(\hat{\delta}_{ij}(x) + \hat{\delta}_{ik}(x) - \hat{\delta}_{jk}(x))$.
7:     Compute averaged estimate $\hat{\theta}_i(x) = \frac{1}{\binom{m-1}{2}} \sum_{j,k \neq i} \hat{\theta}_i^{jk}(x)$.
8: **return** $\hat{\theta}_i(x)$ for all $x \in \mathcal{D}_{\text{test}}, g_i \in G$.

---

different from the true output in terms of Euclidean distance in embedding space. Conversely, if $\theta_i(x)$ is small, we assign larger probability to the setting where $\lambda_i(x)$ and $z^\star(x)$ differ significantly. Finally, note that this graphical model corresponds to a multivariate Gaussian. That is, the vector $[\lambda_1(x) - z^\star(x), \ldots, \lambda_m(x) - z^\star(x)] \in \mathbb{R}^{dm}$ is Gaussian with mean $\mu = \vec{0}$ and a diagonal covariance matrix $\Sigma \in \mathbb{R}^{dm \times dm}$ with $\Sigma_{jj} = \frac{1}{2\theta_{\lceil j/m \rceil}(x)}$. Intuitively, this means that the average difference vector between each $\lambda_i$ and $z^\star(x)$ is centered, with its magnitude inversely proportional to the LLM score $\theta_i(x)$ and independent of other LLMs. Given this probabilistic graphical model, our goal is to learn each quality score $\theta_i(x)$ from the unlabeled test dataset and use these for improved routing.

## 4 Method

Given an unlabeled test dataset $\mathcal{D}_{\text{test}}$ and a pool of LLMs $G$, SMOOTHIE consists of two steps:

1. **Estimation**: The LLM quality scores $\theta_1(x), \ldots, \theta_m(x)$ are learned for each $x \in \mathcal{D}_{\text{test}}$ (Section 4.1, Algorithm 1).

2. **Routing**: The LLM with the highest scores is selected, and its output is used as our final prediction for $x$ (Section 4.2).

We describe each step in the following sections.

### 4.1 LLM score estimation

We describe how to estimate each $\theta_i(x)$s in the graphical model in (1) using only unlabeled data from $\mathcal{D}_{\text{test}}$. Then, we describe how the LLM score estimate can be instantiated to be sample-conditional.

**Computing $\theta_i(x)$**    Below, we state a simple property arising from the fact that (1) corresponds to a multivariate Gaussian with a diagonal covariance matrix.

**Proposition 1** *[85] For any $i, j \in [m]$, it follows from the graphical model in (1) that*

$$\mathbb{E}\left[\|\lambda_i(x) - \lambda_j(x)\|^2\right] = \mathbb{E}\left[\|\lambda_i(x) - z^\star(x)\|^2\right] + \mathbb{E}\left[\|\lambda_j(x) - z^\star(x)\|^2\right]. \tag{2}$$

The proof is in Appendix D and relies on the fact that off-diagonal entries of $\Sigma$ are 0. Note that the left hand side of the equation is observable while the two expectations on the right are unknown. We can apply this equation to pairs of LLM embeddings over a triplet of $\lambda_i, \lambda_j, \lambda_k$ to form a system of three equations with three unknown expectations. Solving, we have

$$\mathbb{E}\left[\|\lambda_i(x) - z^\star(x)\|^2\right] = \frac{1}{2}\left(\delta_{ij}(x) + \delta_{ik}(x) - \delta_{jk}(x)\right) \forall (i, j, k) \in [m], \tag{3}$$

where $\delta_{ij}(x) = \mathbb{E}\left[\|\lambda_i(x) - \lambda_j(x)\|^2\right]$. Since (1) is a multivariate Gaussian with $\Sigma_{jj} = \frac{1}{2\theta_{\lceil j/m \rceil}(x)}$, we can write $\theta_i(x)$ as the following function of $\mathbb{E}\left[\|\lambda_i(x) - z^\star(x)\|^2\right]$:

$$\mathbb{E}\left[\|\lambda_i(x) - z^\star(x)\|^2\right] = \sum_{j=1}^{d} \mathbb{E}\left[(\lambda_{i,j}(x) - z_j^\star(x))^2\right] = \sum_{j=1}^{d} \text{Var}\left(\lambda_{i,j}(x) - z_j^\star(x)\right) = \frac{d}{2\theta_i(x)}, \tag{4}$$

where $\lambda_{i,j}(x)$ and $z_j^\star(x)$ are the $j$th indices of the embeddings $\lambda_i(x)$ and $z^\star(x)$ respectively. Therefore, we can write $\theta_i^{jk}(x) = \frac{d}{\delta_{ij}(x) + \delta_{ik}(x) - \delta_{jk}(x)}$, where each $\delta_{ij}(x)$ can be estimated using the LLM outputs on $\mathcal{D}_{\text{test}}$, and in practice in Algorithm 1 we estimate $\theta_i(x)$ by averaging $\theta_i^{jk}(x)$ over all $\binom{m-1}{2}$ pairs of $j, k \neq i$.

**Sample-conditional estimation of** $\theta_i(x)$ Note that the expectation in $\delta_{ij}(x) = \mathbb{E}\left[\|\lambda_i(x) - \lambda_j(x)\|\right]$ is over the randomness in $\lambda_i(x), \lambda_j(x)$ conditioned on a fixed point $x$. However, we only have one sample per $x$. One simple approach is to use the entire dataset to estimate $\theta_i(x)$, i.e., $\hat{\delta}_{ij}(x) = \frac{1}{n}\sum_{x' \in \mathcal{D}_{\text{test}}} \|\lambda_i(x') - \lambda_j(x')\|^2$. We denote this as SMOOTHIE-GLOBAL. However, in SMOOTHIE-GLOBAL each $\theta_i(x)$ for $i \in [m]$ is a constant over the entire $\mathcal{D}_{\text{test}}$. Therefore, we use nearest neighbor kernel smoothing to estimate each $\delta_{ij}(x)$ in a sample-dependent manner, an approach we call SMOOTHIE-LOCAL. Concretely, for $x \in \mathcal{D}_{\text{test}}$, define $\text{NN}_{n_0}(x) \subset \mathcal{D}_{\text{test}}$ as the $n_0 < n$ nearest neighbors of $x$ (excluding $x$ itself) in $f_0$'s embedding space. Then, we construct $\hat{\delta}_{ij}(x) = \frac{1}{n_0}\sum_{x' \in \text{NN}_{n_0}(x)} \|\lambda_i(x') - \lambda_j(x')\|^2$, and do the same for $\hat{\delta}_{ik}(x), \hat{\delta}_{jk}(x)$ to get a sample-conditional estimate of $\theta_i(x)$. The procedure for estimating $\theta_i(x)$ in SMOOTHIE-LOCAL is outlined in Algorithm 1.

## 4.2 Routing

Once we have estimates of $\hat{\theta}_i(x)$ for each of the $m$ generators by using Algorithm 1, we can construct our route() function. We define $\text{route}(\mathcal{G}, x) = g_i$ where $i = \arg\max\{\theta_1(x), \ldots, \theta_m(x)\}$, which selects the highest scoring LLM for input $x$ based on $\hat{\theta}_i(x)$. We apply this on $\mathcal{D}_{\text{test}}$ to determine the best LLM for each input sample.

## 5 Results

We empirically analyze SMOOTHIE-GLOBAL and SMOOTHIE-LOCAL, focusing on four questions:

1. How well does SMOOTHIE-GLOBAL recover ground-truth LLM rankings over samples belonging to the same task (Section 5.1)?
2. In multi-task datasets, how well can SMOOTHIE-LOCAL perform unsupervised-routing, by identifying the best LLM for each sample (Section 5.2)?
3. Can SMOOTHIE-GLOBAL and SMOOTHIE-LOCAL be applied to select from or route between different prompts (Section 5.3)?
4. How does SMOOTHIE-GLOBAL and SMOOTHIE-LOCAL's performance change as a function of different algorithmic choices (Section 5.4)?

### 5.1 Single-Task LLM Scoring

**Setup** We begin by evaluating whether SMOOTHIE-GLOBAL can accurately learn the relative performance of different LLMs on a single task-dataset. We study three categories of tasks. First, we consider 7 datasets corresponding to commonly-studied natural language generation (NLG) tasks [51]: CNN/DailyMail and XSum (summarization), SQuAD (reading comprehension), TriviaQA (factual recall), E2E and WebNLG (data-to-text generation), and LegalBench's Definition Extraction (text extraction) [1, 27, 30, 31, 43, 61, 62, 70, 81, 84]. We report Rouge2 for summarization and data-to-text generation tasks and accuracy for all others. For all tasks other than Definition Extraction we evaluate SMOOTHIE-GLOBAL on a 1000 sample subset.[2] For these tasks, we consider two ensembles

---
[2]Definition Extraction has fewer than 1000 samples.

of LLMs at different size points. At the 3B size point, our ensemble consists of Pythia-2.8B [7], Gemma-2B [91], Incite-3B [17], and Dolly-3B [18]. At the 7B size point, our ensemble consists of Llama-2 [92], Mistral [40], Vicuna [107], Gemma-7B [91], and Nous Capybara [19]. We manually write a single prompt template for each task, and all model generations rely on this template.

Second, we consider two instruction-following benchmarks: AlpacaEval and MixInstruct [23, 24, 41, 50]. For AlpacaEval, we rely on responses accessible via the online leaderboard.[3] We identify 10 LLMs (each from a different base family), and download these models' responses to the AlpacaEval instructions. We conduct 10 different simulations, where in each simulation we randomly select 5 LLMs from our pool to function as an ensemble. Reported win-rates use the standard GPT-4 references. For MixInstruct, we use generations from an ensemble of 11 different LLMs originally studied in [41]. Following [41], we measure generation quality using a ChatGPT-based rank.

Finally, we consider a more "reasoning-intensive" task, GSM8K [16]. We consider an ensemble of three models: Gemma-7B, Phi-2 [39], and Llema-7b [4]. We prompt each model to provide a chain-of-though reasoning [100], and apply SMOOTHIE to these generations.

For all datasets, we apply SMOOTHIE-GLOBAL using SentenceBERT (`all-mpnet-base-v2`) embeddings of generations [76].

**Results** We first measure how frequently the highest-weighted LLM according to SMOOTHIE-GLOBAL corresponds to the best-performing LLM in the ensemble. We observe that SMOOTHIE-GLOBAL selects the best-performing LLM for 4/7 tasks on the 3B ensemble, and for 5/7 tasks on the 7B ensemble (Figure 8). On AlpacaEval, SMOOTHIE-GLOBAL selects the best-performing LLM by win-rate for 8/10 ensembles, and the best performing LLM by length-controlled win-rate for 7/10 ensembles. On MixInstruct and GSM8K, SMOOTHIE-GLOBAL again identifies the best-performing LLM in the ensemble.

Second, we measure how well SMOOTHIE-GLOBAL captures quality differences between LLMs in the ensemble, by computing the Spearman's rank correlation coefficient between $\theta_i$ and ground truth quality scores ensemble models. Overall, we find that SMOOTHIE-GLOBAL's learned weights approximate the relative ordering of model quality well. On the NLG tasks SMOOTHIE-GLOBAL we measure an average correlation coefficient (across both ensembles and seven tasks) of 0.72. Figure 2(a) visually depicts the distribution of task coefficients—on only one ensemble/dataset pair is there a correlation coefficient $\leq 0$. On MixInstruct, we observe a correlation coefficient of 0.94, and on AlpacaEval, we observe a correlation coefficient of 0.46.

Finally, we measure how the performance of the LLM selected by SMOOTHIE compares to other selection algorithms. We first compare SMOOTHIE-GLOBAL to an unsupervised random baseline (RANDOM), which would select a random model from the ensemble. We reported the *expected* performance of this method, which is equivalent to taking the average performance of the ensemble. We also compare SMOOTHIE-GLOBAL to a labeled baseline which simulates selecting an LLM on the basis of a small amount of validation data [67] (BEST-ON-VAL). We sample a small labeled validation set (50 samples) and select the LLM that performs the best on this set. To account for sampling variation, we repeat this with 10 random draws and report the average performance. Because AlpacaEval has no training split and MixInstruct has no labeled data, we only compare SMOOTHIE-GLOBAL to RANDOM on those datasets.

Table 1 provides results for the seven NLG tasks. We find that SMOOTHIE-GLOBAL outperforms the unsupervised RANDOM baseline on 6/7 tasks for the 3B ensemble and on 7/7 tasks for the 7B ensemble. SMOOTHIE-GLOBAL outperforms RANDOM by up to 7pts (on tasks measured by rouge2), and by up to 12pts (on tasks measured by accuracy). We also observe that SMOOTHIE-GLOBAL is frequently competitive with and even outperforms the BEST-ON-VAL baseline, which uses labeled data. SMOOTHIE-GLOBAL outperforms BEST-ON-VAL on 4/7 tasks for the 3B ensemble, and 5/7 tasks for the 7B ensemble. On GSM8K, SMOOTHIE-GLOBAL achieves a solve-rate of 37.5% (matching BEST-ON-VAL, while RANDOM achieves a solve-rate of 28.3% (Table 11).

SMOOTHIE-GLOBAL also outperforms the RANDOM baseline on the instruction-following datasets. On MixInstruct, SMOOTHIE-GLOBAL achieves a ChatGPT-rank ($\downarrow$) of 3.91, while RANDOM achieves a ChatGPT-rank of 5.95 (Table 10). On AlpacaEval, SMOOTHIE-GLOBAL outperforms RANDOM on all but one trial (across both win-rate and length-controlled win-rate). SMOOTHIE-GLOBAL

---

[3]Responses are available on the AlpacaEval website: https://tatsu-lab.github.io/alpaca_eval/.

|     |                   | CNN  | Def. Ext. | E2E  | SQuAD | TriviaQA | WebNLG | XSum |
|-----|-------------------|------|-----------|------|-------|----------|--------|------|
|     | RANDOM            | 12.9 | 52.4      | 27.3 | 59.6  | 32.7 | 23.4 | 4.5 |
| 3B  | SMOOTHIE-GLOBAL   | **14.3** | **61.5** | **31.8** | 60.7 | 32.1 | **30.7** | 4.5 |
|     | BEST-ON-VAL       | 13.0 | 60.5      | 31.1 | **66.4** | **38.7** | 30.3 | **5.3** |
|     | RANDOM            | 13.7 | 58.5      | 35.3 | 67.9  | 59.3     | 44.1   | 6.9  |
| 7B  | SMOOTHIE-GLOBAL   | **14.5** | **70.9** | **36.9** | **76.2** | **68.3** | 45.9 | **8.4** |
|     | BEST-ON-VAL       | **14.5** | 69.4  | 36.7 | 74.0  | 65.8     | **48.3** | 8.3  |

**Table 1:** Comparing SMOOTHIE-GLOBAL to baseline methods on different ensembles across NLG datasets. Underlined values are the best performing *unsupervised* methods. Bold values are the best performing *overall* methods. We report rouge2 scores for CNN, XSum, WebNLG, and E2E, and accuracy for the rest. All metrics are scaled to 0-100.

outperforms RANDOM by an average of 15pt win-rate, and up to 27pts. Figure 2(b) and Figure 2(c) visualize this distribution.

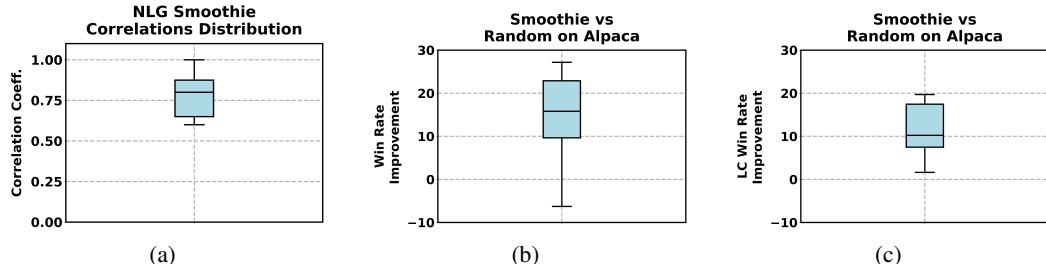

(a)               (b)               (c)

**Figure 2: (a)** Spearman's rank correlation coefficient between SMOOTHIE-GLOBAL weights and ground-truth LLM performance for 3B and 7B ensembles across NLG tasks. **(b)** SMOOTHIE-GLOBAL's improvement over RANDOM by win-rate on AlpacaEval. (c)SMOOTHIE-GLOBAL's improvement over RANDOM by length-controlled win-rate on AlpacaEval.

## 5.2 Multi-task Routing

**Setup** We next assess whether SMOOTHIE-LOCAL's sample-conditional scoring mechanism allows it to route samples to LLMs in the multi-capability regime. We construct two mixed-task distributions by combining existing datasets. The first distribution corresponds to tasks measured by accuracy, and contains SQuAD, TriviaQA, and Definition Extraction. We refer to this as DISTR-ACC. The second distribution corresponds to tasks measured by Rouge2, and contains CNN/DailyMail, XSum, Web NLG, and E2E. We refer to this as DISTR-ROUGE2. For each mixed-task dataset, we report the metric averaged across all tasks. We compare to three baselines.

- RANDOM: A random-selection baseline which returns a generation from a random LLM in the ensemble. Though naive, prior work has found this to be a strong method in practice [56]. We run 10 trials and report the mean of this approach to account for variance.
- LABELED-KNN: A labeled data-based KNN baseline. For this, we sample 50 labeled samples from a separate hold-out set ($\mathcal{D}_{val}$), and measure the performance of each candidate LLM on this set. For a given test sample $x$, we identify the 20 most semantically similar instances in $\mathcal{D}_{val}$ (using SentenceBERT embeddings [76]), and route $x$ to the highest performing LLM over this subset. We note that the LABELED-KNN baseline is derived from routing methods in [48, 86].
- PAIRRM: A reward model from [41] which accepts an instruction and multiple generations as input, scores each generations suitability for the instruction, and returns the predicted best generation. PAIRRM is a labeled-data method which [41] trained on collected preference data.

In addition, we also compare the best individual model in the ensemble (BEST-MODEL), and SMOOTHIE-GLOBAL. For both mixed-task datasets, we run SMOOTHIE-LOCAL with SentenceBERT

|        | 3B | | 7B | |
|--------|------------|---------------|------------|---------------|
| **Method** | DISTR-ACC | DISTR-ROUGE2 | DISTR-ACC | DISTR-ROUGE2 |
| RANDOM | 48.7 | 17.0 | 65.4 | 25.0 |
| PAIRRM | 53.9 | 19.0 | 71.8 | 25.5 |
| LABELED-KNN | 51.0 | 16.8 | 71.7 | 26.2 |
| BEST-MODEL | 52.3 | 18.1 | 73.2 | 26.4 |
| SMOOTHIE-GLOBAL | 51.3 | 18.1 | 66.5 | 26.1 |
| SMOOTHIE-LOCAL | **58.7** | **20.2** | **75.0** | **26.9** |

**Table 2:** Comparing SMOOTHIE-LOCAL to baseline methods on the 3B and 7B ensembles for multi-task distributions. DISTR-ACC and DISTR-ROUGE2 are measured with accuracy and rouge2 respectively. Bold values indicate the best performing method for each dataset and model size. Metrics are scaled to 0-100.

embeddings, and the sample-conditional version of SMOOTHIE-LOCAL estimates $\theta_i(x)$ using a neighborhood size $n_0 = 1$.

Results for the 3B and 7B ensembles over DISTR-ACC and DISTR-ROUGE2 are provided in Table 2. We find that SMOOTHIE-LOCAL outperforms all baselines across both data distributions, for both ensembles. Though SMOOTHIE-LOCAL requires no labels, it still outperforms labeled data baselines like LABELED-KNN and PAIRRM. We observe a substantial gap between SMOOTHIE-LOCAL and SMOOTHIE-GLOBAL, which indicates that SMOOTHIE-LOCAL's sample-specific scoring mechanism provides performance improvements.

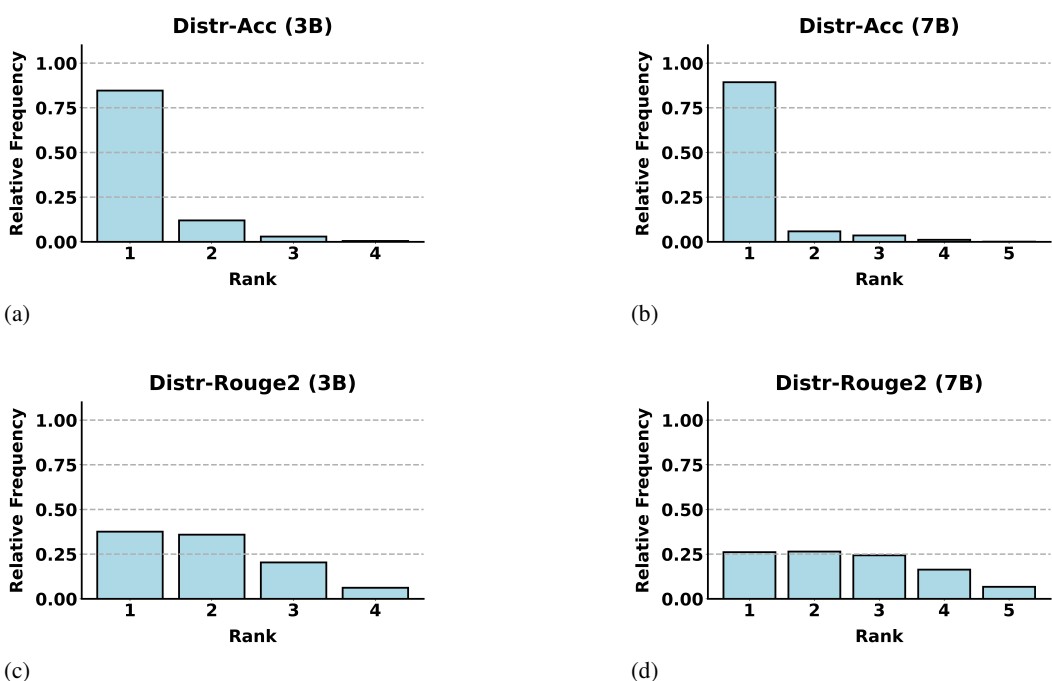

(a)                     (b)

(c)                     (d)

**Figure 3:** On DISTR-ACC and DISTR-ROUGE2, we measure how frequently SMOOTHIE-LOCAL selects the $i$-th best generation across the ensemble, for both the 3B and 7B ensembles.

Notably, we see that SMOOTHIE-LOCAL substantially betters BEST-MODEL, indicating that SMOOTHIE-LOCAL's routing mechanism is offering a performance improvement over a strategy which merely selects the best LLM on average. We study this in greater detail by examining the relative rank of the LLM selected by SMOOTHIE-LOCAL for each sample. For each sample in

|  |  | CNN | Def. Ext. | E2E | SQuAD | TriviaQA | WebNLG | XSum |
|---|---|---|---|---|---|---|---|---|
| Falcon | RANDOM | 7.1 | 60.3 | 27.8 | 47.3 | 22.0 | _29.2_ | 4.7 |
|  | SMOOTHIE-GLOBAL | _7.9_ | _62.2_ | **31.6** | **53.3** | **31.4** | 28.3 | _6.4_ |
|  | SMOOTHIE-LOCAL | 8.0 | **69.2** | 31.5 | **53.3** | 27.4 | 30.8 | 6.0 |
|  | BEST-ON-VAL | **8.4** | 64.2 | 31.0 | 52.7 | **31.4** | **32.5** | **6.7** |
| Llama-2 | RANDOM | _7.3_ | 47.8 | 31.6 | 54.0 | 45.9 | 45.5 | 11.2 |
|  | SMOOTHIE-GLOBAL | 6.9 | **64.6** | **37.6** | _61.4_ | **68.7** | _48.5_ | _12.8_ |
|  | SMOOTHIE-LOCAL | 9.5 | 59.3 | 33.6 | 63.1 | 61.3 | 48.0 | 12.7 |
|  | BEST-ON-VAL | **11.8** | **64.6** | 35.0 | **66.1** | **68.7** | **48.7** | **13.0** |

**Table 3:** Comparing SMOOTHIE-GLOBAL and SMOOTHIE-LOCAL to baseline methods in the prompt-selection setting. Underlined values are the best performing *unsupervised* methods. Bold values are the best performing *overall* methods. We report rouge2 scores for CNN, XSum, WebNLG, and E2E, and accuracy for the rest. All metrics are scaled to 0-100.

DISTR-ACC and DISTR-ROUGE2, we rank the quality of each LLM's generation according to standard-competition ranking (i.e., "1-2-2-4" ranking). We then count how frequently SMOOTHIE-LOCAL selects the rank-$i$ generation across each distribution for each ensemble. We visualize results in Figure 3. As the visualizations demonstrate, SMOOTHIE-LOCAL consistently selects the best or second-best generation from within the ensemble.

### 5.3 Prompt Selection

Third, we study whether SMOOTHIE-LOCAL and SMOOTHIE-GLOBAL can be generalized to other settings where engineers have a candidate pool of text generators of unknown quality, and must select one of them to use for some application. In particular, we focus on the setting where an engineer has access to multiple prompt templates for a given generation task, and must select which prompt-templates' generation to use as the final output [29]. Unlike above, we assume the engineer only has access to one LLM. We study SMOOTHIE-LOCAL and SMOOTHIE-GLOBAL in this regime using the NLG tasks from Section 5.1. For each task, we manually write between 3 and 5 prompt templates, varying the wording of instructions and the choice of in-context samples. We analyze SMOOTHIE applied to two models at different size points: Falcon (1B) [65] and Llama-2 (7B) [92].

Table 3 provides the results. Overall, we find that SMOOTHIE-GLOBAL selects the optimal prompt 2/7 times for Falcon-1B, and 3/7 times for Llama-2. SMOOTHIE-LOCAL and SMOOTHIE-GLOBAL consistently outperform RANDOM–on 6/7 tasks for Falcon-1b and 6/7 tasks for Llama-2. On 7 task/model combinations, one of either SMOOTHIE-GLOBAL or SMOOTHIE-LOCAL matches or outperforms a labeled baseline. To better contextualize performance improvements from SMOOTHIE-GLOBAL, we also compare to the improvement that accompanies increasing model size. Following a common practice in recent work, we can quantify the extent to which SMOOTHIE-GLOBAL allows smaller models to match or exceed the performance of larger models [2, 29]. In Figure 4 (Appendix E), we compare RANDOM and SMOOTHIE-GLOBAL on models from the Pythia suite at four sizes: 410M, 1B, 2.8B, and 6.9B parameters [7]. We observe that SMOOTHIE-GLOBAL substantially improves performance—on E2E, SMOOTHIE-GLOBAL enables a 410M parameter model to outperform a 6.9B parameter model.

### 5.4 Ablations

Finally, we conduct ablations to examine different aspects of SMOOTHIE-GLOBAL and SMOOTHIE-LOCAL: improving its efficiency, adjusting the neighborhood size, varying the choice of embedding model, and using different LLM ensembles.

**Improving efficiency** First, we explain SMOOTHIE's current efficiency properties. To estimate the Smoothie weights for routing, we use a simple closed-form procedure that does not require any SGD or training, as described in Algorithm 1. As a result, SMOOTHIE weights on the entire dataset can be computed in seconds—for the 7B ensemble, SMOOTHIE-LOCAL on the multi-task datasets takes

2.14 seconds per 1000 samples, and SMOOTHIE-GLOBAL on the single-task datasets takes under 0.03 seconds per 1000 samples. Moreover, SMOOTHIE does not require any ground-truth annotations; however, all $m$ model generations per test sample are needed as input to the algorithm. That is, we need $n \times m$ generations for a $\mathcal{D}_{\text{train}}$ of size $n$ samples.

Fortunately, the need for computing all model generations per test sample can be removed with a small algorithm tweak, making Smoothie even more efficient and its runtime independent of $n$. Suppose we have a held-out set of $n_{\text{train}}$ train samples with precomputed generations from the models in the ensemble. For each test sample, we retrieve the most similar train samples, learn the Smoothie weights for the sample using the corresponding train sample generations, and return the model with the highest Smoothie weight (i.e., in line 5 in Algorithm 1, KNN is now over a held-out training dataset). This approach, which we call SMOOTHIE-TRAIN, selects the model for a test sample without needing model generations for that sample. Only $n_{\text{train}} \times m$ generations are needed, regardless of how large the test dataset $n$ is.

We study the NLG tasks, using $n_{\text{train}} = 250$ samples. In Table 7 (Appendix E), we evaluate a version of SMOOTHIE-GLOBAL-TRAIN) and observe that it matches SMOOTHIE-GLOBAL on 12/14 model-dataset pairs, and performs worse on the remaining 2/14 pairs. We also evaluate SMOOTHIE-LOCAL-TRAIN, on DISTR-ACC and DISTR-ROUGE2 (Table 8) using a neighborhood of size $n_0 = 20$. We find here that while SMOOTHIE-LOCAL-TRAIN underperforms SMOOTHIE-LOCAL on both the 3B and 7B ensemble for both DISTR-ACC and DISTR-ROUGE2, it still outperforms RANDOM and remains competitive with supervised baselines.

**Neighborhood size** We study the impact of $n_0$, and consider SMOOTHIE-LOCAL's performance for $n_0 \in [1, 5, 10, 20, 50, 100]$. Figure 5 provides performance over DISTR-ACC and Figure 6 provides performance over DISTR-ROUGE2. Overall, we find that SMOOTHIE-LOCAL's performance steadily degrades as $n_0$ increases, and is highest when $n_0 = 1$.

**Choice of embeddings** We study how the choice of embeddings affects SMOOTHIE-GLOBAL performance (Table 9). Specifically, we compare the performance of SMOOTHIE-LOCAL using Sentence-Bert embeddings (`all-mpnet-base-v2`) [76] to BGE embeddings (`bge-small-en-v1.5`) [102]. We observe that SMOOTHIE-LOCAL appears robust to different embeddings—SMOOTHIE-LOCAL with BGE embeddings still outperforms other labeled and unlabeled baselines. Interestingly, we observe that certain embedding models appear to yield better performance over certain distribution/ensemble combinations. For instance, SMOOTHIE-LOCAL with SentenceBERT embeddings outperforms SMOOTHIE-LOCAL with BGE embeddings on DISTR-ACC for the 3B ensemble and DISTR-ROUGE2 for the 7B ensemble, while performing worse on DISTR-ROUGE2 for the 3B ensemble and DISTR-ACC for the 7B ensemble.

**Different ensembles** Finally, we consider whether SMOOTHIE-GLOBAL can generalize to a wider array of ensembles (Figure 7). We combine the LLMs contained in the 3B and 7B ensembles into a single pool, and sample 50 distinct ensembles ranging in size from 4-7 LLMs. For each of the 7 NLG tasks, we evaluate SMOOTHIE-GLOBAL's ability to identify the best model from within each ensemble. Across these 350 settings, we find that SMOOTHIE-GLOBAL identifies the best model in 211 of them (60.2% of the time), and one of the two best models in 292 of them (83% of the time).

# 6    Conclusion

In this paper we study and propose an algorithm for learning label-free routers for generative tasks. We validate our approach across a variety of evaluation regimes, finding it consistently beats other unsupervised approaches and often matches/exceeds supervised approaches.

**Limitations** We discuss several of SMOOTHIE's limitations. First, its multivariate Gaussian graphical model currently uses a diagonal covariance matrix. This assumes independent error vectors for each generation, though SMOOTHIE could be extended to account for dependencies [72, 93]. Additionally, SMOOTHIE optimizes only for performance without considering cost tradeoffs between large and small models. Finally, its reliance on embeddings may capture only certain aspects of semantic similarity. Other embedding models and additional heuristics could be used to create richer input features for SMOOTHIE.

# 7 Acknowledgements

We thank the Microsoft Accelerate Foundation Models Research Program for providing portions of the compute used for the results in this paper. We thank Gautam Machiraju, Jon Saad-Falcon, Krista Opsahl-Ong, Sabri Eyuboglu, Jordan Juravsky, Vishnu Sarukkai, Ben Spector, Eric Nguyen, Jerry Liu, Chris Fifty, Avanika Narayan, and Michael Zhang for their helpful feedback and discussion.

We gratefully acknowledge the support of NIH under No. U54EB020405 (Mobilize), NSF under Nos. CCF2247015 (Hardware-Aware), CCF1763315 (Beyond Sparsity), CCF1563078 (Volume to Velocity), and 1937301 (RTML); US DEVCOM ARL under Nos. W911NF-23-2-0184 (Long-context) and W911NF-21-2-0251 (Interactive Human-AI Teaming); ONR under Nos. N000142312633 (Deep Signal Processing); Stanford HAI under No. 247183; NXP, Xilinx, LETI-CEA, Intel, IBM, Microsoft, NEC, Toshiba, TSMC, ARM, Hitachi, BASF, Accenture, Ericsson, Qualcomm, Analog Devices, Google Cloud, Salesforce, Total, the HAI-GCP Cloud Credits for Research program, the Stanford Data Science Initiative (SDSI), and members of the Stanford DAWN project: Meta, Google, and VMWare. NG is suported by the Stanford Interdisciplinary Graduate Fellowship (SIGF). The U.S. Government is authorized to reproduce and distribute reprints for Governmental purposes notwithstanding any copyright notation thereon. Any opinions, findings, and conclusions or recommendations expressed in this material are those of the authors and do not necessarily reflect the views, policies, or endorsements, either expressed or implied, of NIH, ONR, or the U.S. Government.

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

# A  Appendix

In Appendix B, we provide a glossary of notation used in the paper. In Appendix C, we provide an extended related work, and in Appendix D we provide a proof of Proposition 1, which is used in deriving the SMOOTHIE algorithm. Finally, in Appendix E we provide additional experimental results and details.

Code for reproducing our results and using SMOOTHIE is available at `https://github.com/HazyResearch/smoothie`.

# B  Notation

The glossary is given in Table 4 below.

| Symbol | Used for |
|---|---|
| $\bar{\mathcal{V}}$ | The space of all vocabulary sequences. |
| $x$ | Input text $x \in \mathcal{X} \subset \bar{\mathcal{V}}$. |
| $y$ | Reference output text $y \in \mathcal{Y} \subset \bar{\mathcal{V}}$. |
| $G$ | Candidate pool of $m$ LLMs, $G = \{g_1, \ldots, g_m\}$, where each $g_i \in \mathcal{G} : \mathcal{X} \to \mathcal{Y}$ produces a generation $g_i(x)$ on input $x$. |
| $\mathcal{D}_{\text{test}}$ | Unlabeled test dataset $\mathcal{D}_{\text{test}} = \{x_i\}_{i=1}^n$. |
| route | Routing function route $: \mathcal{G}^m \times \mathcal{X} \to \mathcal{G}$ that selects the best LLM from $G$ for each sample. |
| $\theta_i(x)$ | Quality score of the $i$th LLM on test sample $x$, also used in the graphical model in (1). |
| $z_{g_0}$ | Embedding mapping $z_{g_0} : \bar{\mathcal{V}} \to \mathbb{R}^d$ for any text sequence, where $g_0$ is an embedding model such as SentenceBERT [76]. |
| $\lambda_i(x)$ | The observable embedding of $x$ concatenated with the $i$th LLM's generated output, $\lambda_i(x) := z_{g_0}([x, g_i(x)])$. |
| $z^\star(x)$ | The latent embedding of $x$ concatenated with unknown reference output, $z^\star(x) := z_{g_0}([x, y])$. |
| $Z$ | Partition function for normalization of (1). |
| $n_0$ | Number of nearest neighbors used to learn $\theta_i(x)$ for $x$. $n_0 = n$ (i.e., the entire test dataset) corresponds to SMOOTHIE-GLOBAL and $n_0 < n$ corresponds to SMOOTHIE-LOCAL. |
| $\hat{\delta}_{ij}(x)$ | The average squared Euclidean distance between the $i$th and $j$th LLM embeddings over a neighborhood around $x$, $\hat{\delta}_{ij}(x) = \frac{1}{n_0} \sum_{x' \in \text{NN}_{n_0}(x)} \|\lambda_i(x') - \lambda_j(x')\|^2$. This is the primary expression used in computing $\theta_i(x)$. |

**Table 4:** Glossary of variables and symbols used in this paper.

# C  Extended Related Work

**LLM Routing** The problem of determining how to route samples to various models has been long studied in statistics [37, 42] as well as Mixture of Experts deep neural networks [25, 82]. These works focus on how to jointly train the models and router in a stable and efficient manner.

Since many LLMs are now available off-the-shelf, recent works study how routing mechanisms can be applied at inference time to trained models. Some works involve training task or domain-specific expert models and then learning a router. The router can be a nearest neighbors algorithm [38], a neural network [96] that classifies among the different domains corresponding to the experts, or an extra gate learned when training the expert models [60]. These approaches do not explicitly require labels, but they require knowledge of what domain is used to train each expert and assume that each expert is the best model for its corresponding domain, therefore effectively using this mapping as a form of labels. In contrast, our setting focuses on routing among pre-trained LLMs where we do not know what models are optimal on what tasks and their samples.

A second category of inference-time routing works studies how to choose among a collection of pre-trained LLMs, which is the setting that SMOOTHIE focuses on. Several approaches involve training a meta-model that either scores or ranks how a LLM will perform on a sample [41, 74, 79], all of which required labeled data to train. MoRE [87] involves training a simpler random forest classifier, using the rate of agreement among LLMs as one of the features, which is similar to how Smoothie estimates scores; however, it also requires labeled data to train the classifier. Some approaches [48, 86] do not

require training routers and simply use nearest neighbor methods. However, these nearest neighbor methods still use labeled data to determine what training samples each LLM performs the best on. [55] invokes a trained reward model for the routing mechanism. [21] trains a classification-based router using the BARTScore metric on LLM generations as pseudolabels; this avoids using manually labeled data, demonstrating that while a majority of routing methods require labeled data, there exist some alternatives that do not. We leave it to future work to compare and integrate SMOOTHIE with other unsupervised approaches.

Finally, complementary to our setting are works that jointly focus on cost minimization as well as quality of generations. RouterBench [33] creates a benchmark for studying the cost-quality tradeoffs in routing systems. Optimizing for cost can be done algorithmically, such as in FrugalGPT [13], AutoMix [58], RouteLLM [63], and [88], as well as via hardware enhancements such as SambaNova Systems' Composition of Experts [69].

**LLM Ensembling** A rich literature has observed that ensembling LLM outputs—across different prompts or base models—can improve the accuracy of generated predictions. Prior work has proposed and studied a number of different ensembling algorithms for classification tasks, including majority-voting [54, 98], weak-supervision [2, 29], boosting [32, 68, 104], and others [49, 66, 80].

More relevant to our work here is a literature on ensembling for generative tasks. One category of methods rely on an auxilliary sequence-to-sequence models to "fuse" generations from different prompts or base LLMs [41]. Though recently applied in the context of modern LLMs, the concept of fusion traces back to older work on summarization [5, 46, 47, 75]. Some techniques combine or switch among multiple outputs at inference time [34, 36, 59, 83, 97], while others involve averaging in weight space [35, 38, 95]. Lastly, ensembling can also be approximated by randomly selecting a model to be used in multi-turn settings [56].

**Other LLM Selection Algorithms** Beyond the setting of selecting among multiple LLMs, other works have explored how to select the optimal prompt template from a collection of candidate prompts. These works can be grouped into two categories. The first category assumes that engineers have access to labeled data. In the naive case, this labeled data can simply be used to select the best performing prompt [45, 67, 77]. Another subset of this category focuses on the setting where new prompts can be generated by selecting in-context demonstrations from a set of labeled samples (typically a small training set) [22, 53]. Prior work has proposed different methods for identifying the optimal in-context demonstrations to use, depending on the sample for which the LLM is being used to produce a prediction for [11, 77, 90, 101, 105, 106]. The second category focuses on zero-label prompt selection methods, but solely for classification tasks [52, 89, 103]. Prior work here selects prompts on the basis of mutual information [89], agreement rates between predictions produced by different prompts [52], and various probability based measures [28, 57, 103].

**Weak supervision** SMOOTHIE utilizes techniques inspired by weak supervision literature. Weak supervision aims to programmatically generate labels on an unlabeled dataset by aggregating the predictions of several weak "voters", such as heuristics, pre-trained models, and knowledge graphs [71, 73]. It assumes a particular latent variable graphical model and uses its structure to estimate latent quantities, such as the accuracy of each voter (in our setting, the quality score of each LLM). Typically, this graphical model is a binary Ising model, as weak supervision has generally been studied in classification settings [26, 72], where embeddings have been utilized as auxiliary signal but not modeled explicitly [15, 29]. Weak supervision has been applied to broader settings, such as for learning rankings, graphs, and manifolds [85, 94]. We derive our estimation procedure from the Gaussian model in [85], applying it to LLM embeddings. While both SMOOTHIE and [85] use a multivariate Gaussian model, in SMOOTHIE we apply it to model routing with SBERT embeddings on natural language datasets, whereas [85] conducts synthetic experiments in hyperbolic spaces and metric spaces induced by synthetic graphs. Moreover, SMOOTHIE uses nearest neighbor kernel smoothing to allow for sample-dependent weights—critical for routing—while [85] calculates one global set of weights over the dataset.

**Consistency-based selection** Consistency is central to unsupervised selection and aggregation methods, the simplest being majority vote. While weak supervision methods [26] and SMOOTHIE heavily rely on notions of voter agreement as depicted in a graphical model, there are several other consistency-based methods. Minimum Bayes Risk methods [6, 44] selects the generation that has the highest average similarity (i.e., cosine) with other generations. This is similar to SMOOTHIE, which routes to the lowest value of (3). If we ignore the subtraction of $\delta_{jk}(x)$ in (3) and average

over more than just $\delta_{ij}(x)$ and $\delta_{ik}(x)$, then SMOOTHIE with $n_0 = 1$ is equivalent to [44] Therefore, SMOOTHIE can be considered as a slightly modified and more general version of this approach. Another approach [10] relies on consistency between a "global" and "local" embedding for each generation. They solve an optimization problem that estimates each generation's quality score by constructing a loss that enforces that the similarity between the estimated true generation (produced by a weighted average of candidates) and the candidate generation should be the same according to both global and local embeddings. In contrast, SMOOTHIE uses one embedding space, relies on a multivariate Gaussian structure among embeddings, and does not require gradient descent to learn the quality of each generation.

**Test-Time Compute** Approaches like model routing, ensembling, and selection can all be seen as ways of utilizing *test-time compute* to produce higher-quality generations from a system of LLMs. Test-time compute can also be utilized over a single LLM via techniques such as those used in OpenAI's o1, Chain of Thought, and Rephrase and Respond [20, 64, 99]. Other works have recently studied how test-time compute scales [9, 12]—finding that producing more generations can often yield the correct response—and how to combine multiple test-time methods, such as Archon [78]. It is interesting future work to consider how SMOOTHIE can be integrated with other test-time compute techniques.

# D  Proof of Proposition 1

We provide a proof of proposition 1, which is a direct property of multivariate Gaussians that is also presented in [85]. We first expand $\mathbb{E}\left[\|\lambda_i(x) - \lambda_j(x)\|^2\right]$:

$$\mathbb{E}\left[\|\lambda_i(x) - \lambda_j(x)\|^2\right] = \mathbb{E}\left[\|(\lambda_i(x) - z^\star(x)) - (\lambda_j(x) - z^\star(x))\|^2\right] \tag{5}$$
$$= \mathbb{E}\left[\|\lambda_i(x) - z^\star(x)\|^2\right] + \mathbb{E}\left[\|\lambda_j(x) - z^\star(x)\|^2\right] - 2\mathbb{E}\left[(\lambda_i(x) - z^\star(x))^\top(\lambda_j(x) - z^\star(x))\right]$$

Let $\lambda_{i,k}(x)$ be the $k$th element of the $\lambda_i(x)$ embedding, and similarly define $z_k^\star(x)$. Note that since $\Sigma$ is diagonal, we can write

$$\mathbf{Cov}\left[\lambda_{i,k}(x) - z_k^\star(x), \lambda_{j,k}(x) - z_k^\star(x)\right]$$
$$= \mathbb{E}\left[(\lambda_{i,k}(x) - z_k^\star(x)) \cdot (\lambda_{j,k}(x) - z_k^\star(x))\right] - \mathbb{E}\left[\lambda_{i,k}(x) - z_k^\star(x)\right]\mathbb{E}\left[\lambda_{j,k}(x) - z_k^\star(x)\right]$$
$$= 0$$

for all $k \in [d]$. Since $\mu = \vec{0}$, we thus have that $\mathbb{E}\left[(\lambda_{i,k}(x) - z_k^\star(x)) \cdot (\lambda_{j,k}(x) - z_k^\star(x))\right] = 0$ for all $k \in [d]$, which implies that $\mathbb{E}\left[(\lambda_i(x) - z^\star(x))^\top(\lambda_j(x) - z^\star(x))\right] = 0$. Plugging this into (5), we have

$$\mathbb{E}\left[\|\lambda_i(x) - \lambda_j(x)\|^2\right] = \mathbb{E}\left[\|\lambda_i(x) - z^\star(x)\|^2\right] + \mathbb{E}\left[\|\lambda_j(x) - z^\star(x)\|^2\right]. \tag{6}$$

# E  Additional Experiments and Details

This section contains additional details on experiments discussed in Section 5.

## E.1  Datasets and Models

Table 5 provides links to the Huggingface datasets used for each task. For E2E, CNN/DailyMail, XSum, and Web NLG we measure performance using rouge2. For SQuAD, TriviaQA, and Definition Extraction we measure using "accuracy." A model generation is treated as "correct" if if contains the answer, and incorrect otherwise [1].

| Dataset name | Huggingface URL |
|---|---|
| E2E | https://huggingface.co/datasets/e2e_nlg |
| CNN/DailyMail | https://huggingface.co/datasets/cnn_dailymail |
| SQuAD | https://huggingface.co/datasets/hazyresearch/based-squad |
| XSum | https://huggingface.co/datasets/EdinburghNLP/xsum |
| TriviaQA | https://huggingface.co/datasets/mandarjoshi/trivia_qa |
| Web NLG | https://huggingface.co/datasets/web_nlg |
| Definition Extraction | https://huggingface.co/datasets/nguha/legalbench |

**Table 5:** Datasets used.

Table 6 contains links for all models used.

| Model name | Huggingface URL |
|---|---|
| Pythia-410M | https://huggingface.co/EleutherAI/pythia-410m |
| Pythia-1B | https://huggingface.co/EleutherAI/pythia-1b |
| Pythia-2.8B | https://huggingface.co/EleutherAI/pythia-2.8b |
| Pythia-6.9B | https://huggingface.co/EleutherAI/pythia-6.9b |
| Gemma-2B | https://huggingface.co/google/gemma-2b-it |
| Incite-3B | https://huggingface.co/togethercomputer/RedPajama-INCITE-Instruct-3B-v1 |
| Dolly-3B | https://huggingface.co/databricks/dolly-v2-3b |
| Llama-2-7B | https://huggingface.co/meta-llama/Llama-2-7b-hf |
| Mistral-7B | https://huggingface.co/mistralai/Mistral-7B-Instruct-v0.2 |
| Vicuna-7B | https://huggingface.co/lmsys/vicuna-7b-v1.5 |
| Gemma-7B | https://huggingface.co/google/gemma-7b |
| Nous Capybara | https://huggingface.co/NousResearch/Nous-Capybara-7B-V1.9 |
| Phi-2 | https://huggingface.co/microsoft/phi-2 |
| Llema-7B | https://huggingface.co/EleutherAI/llemma_7b |

**Table 6:** Huggingface model URLs.

For the Alpaca leaderboard experiments, we run each trial by sampling 5 models from the following set of 10: Nanbeige-Plus-Chat-v0.1, claude-2, Qwen1.5-110B-Chat, yi-large-preview, gemini-pro, Meta-Llama-3-70B-Instruct, Ein-70B-v0.1, mistral-large-2402, Storm-7B, FsfairX-Zephyr-Chat-v0.1.

## E.2 Additional Results

| | | CNN | Def. Ext. | E2E | SQuAD | TriviaQA | WebNLG | XSum |
|---|---|---|---|---|---|---|---|---|
| **3B** | RANDOM | 12.9 | 52.4 | 27.3 | 59.6 | 32.7 | 23.4 | 4.5 |
| | SMOOTHIE-GLOBAL | **14.3** | **61.5** | **31.8** | 60.7 | 32.1 | **30.7** | 4.5 |
| | SMOOTHIE-GLOBAL-TRAIN | 14.3 | 61.5 | 24.7 | 60.7 | 32.1 | **30.7** | 4.5 |
| | BEST-ON-VAL | 13.0 | 60.5 | 31.1 | **66.4** | **38.7** | 30.3 | **5.3** |
| **7B** | RANDOM | 13.7 | 58.5 | 35.3 | 67.9 | 59.3 | 44.1 | 6.9 |
| | SMOOTHIE-GLOBAL | **14.5** | **70.9** | **36.9** | **76.2** | **68.3** | 45.9 | **8.4** |
| | SMOOTHIE-GLOBAL-TRAIN | 14.5 | 70.9 | 36.5 | **76.2** | **68.3** | 45.9 | **8.4** |
| | BEST-ON-VAL | **14.5** | 69.4 | 36.7 | 74.0 | 65.8 | **48.3** | 8.3 |

**Table 7:** We compare SMOOTHIE-GLOBAL to SMOOTHIE-GLOBAL-TRAIN, for which weights are learned on a hold-out set. We provide results from baseline methods for reference. Underlined values are the best performing *unsupervised* methods. Bold values are the best performing *overall* methods. We report rouge2 scores for CNN, XSum, WebNLG, and E2E, and accuracy for the rest. All metrics are scaled to 0-100.

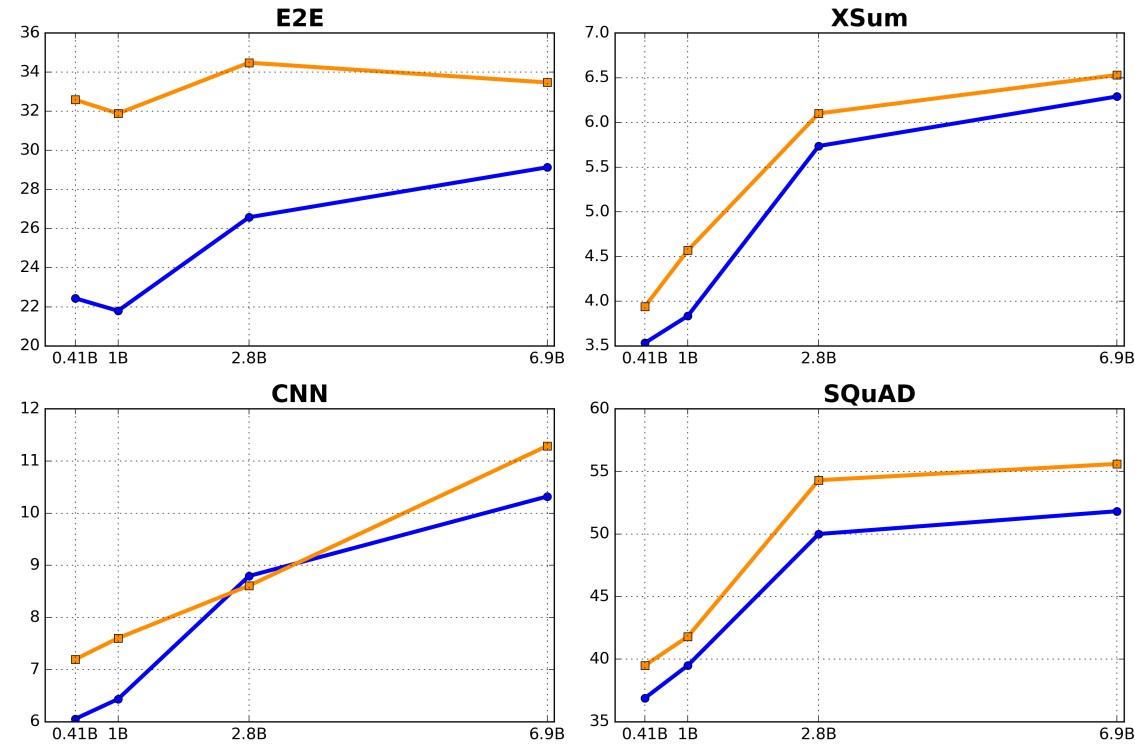

**Figure 4:** We compare RANDOM (blue) and SMOOTHIE-GLOBAL (orange) for prompt-selection on different sized models in the Pythia suite. The x-axis denotes model size, and the y-axis denotes performance (either rouge2 or accuracy).

| Method | 3B | | 7B | |
|---|---|---|---|---|
| | DISTR-ACC | DISTR-ROUGE2 | DISTR-ACC | DISTR-ROUGE2 |
| RANDOM | 48.7 | 17.0 | 65.4 | 25.0 |
| PAIRRM | 53.9 | 19.0 | 71.8 | 25.5 |
| LABELED-KNN | 51.0 | 16.8 | 71.7 | 26.2 |
| BEST-MODEL | 52.3 | 18.1 | 73.2 | 26.4 |
| SMOOTHIE-GLOBAL | 51.3 | 18.1 | 66.5 | 26.1 |
| SMOOTHIE-LOCAL | **58.7** | **20.2** | **75.0** | **26.9** |
| SMOOTHIE-GLOBAL-TRAIN | 51.3 | 18.1 | 66.5 | 26.1 |
| SMOOTHIE-LOCAL-TRAIN | 50.7 | 18.8 | 70.9 | 26.0 |

**Table 8:** We compare SMOOTHIE-LOCAL to SMOOTHIE-LOCAL-train, for which weights are learned on a hold-out set, on the 3B and 7B ensembles for multi-task distributions. DISTR-ACC and DISTR-ROUGE2 are measured with accuracy and rouge2 respectively. Bold values indicate the best performing method for each dataset and model size. Metrics are scaled to 0-100. Other baseline methods are provided for comparison.

|  | 3B | | 7B | |
| --- | --- | --- | --- | --- |
| **Method** | DISTR-ACC | DISTR-ROUGE2 | DISTR-ACC | DISTR-ROUGE2 |
| RANDOM | 48.7 | 17.0 | 65.4 | 25.0 |
| PAIRRM | 53.9 | 19.0 | 71.8 | 25.5 |
| LABELED-KNN | 51.0 | 16.8 | 71.7 | 26.2 |
| BEST-MODEL | 52.3 | 18.1 | 73.2 | 26.4 |
| SMOOTHIE-LOCAL (BGE-small [102]) | **59.3** | 19.7 | 74.6 | **27.1** |
| SMOOTHIE-LOCAL (SBERT [76]) | 58.7 | **20.2** | **75.0** | 26.9 |

**Table 9:** Comparing SMOOTHIE-LOCAL with different embeddings on the 3B and 7B ensembles for multi-task distributions. DISTR-ACC and DISTR-ROUGE2 are measured with accuracy and rouge2 respectively. Bold values indicate the best performing method for each dataset and model size. Metrics are scaled to 0-100.

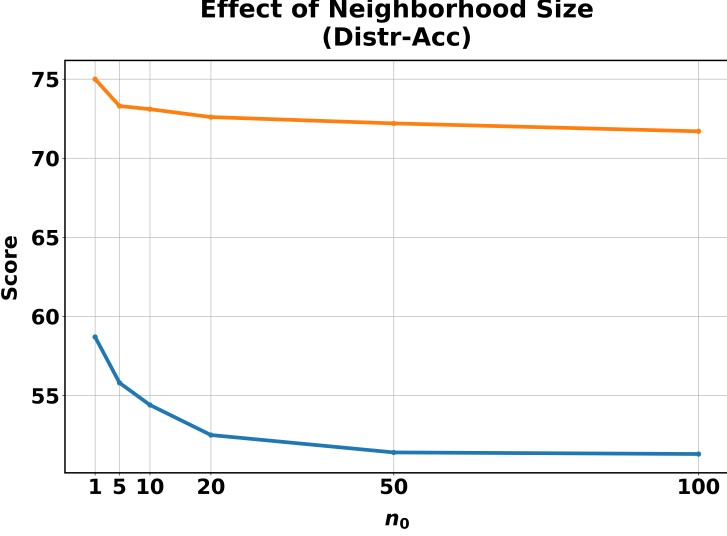

**Figure 5:** We measure how SMOOTHIE-LOCAL's performance on DISTR-ACC changes as $n_0$ changes.

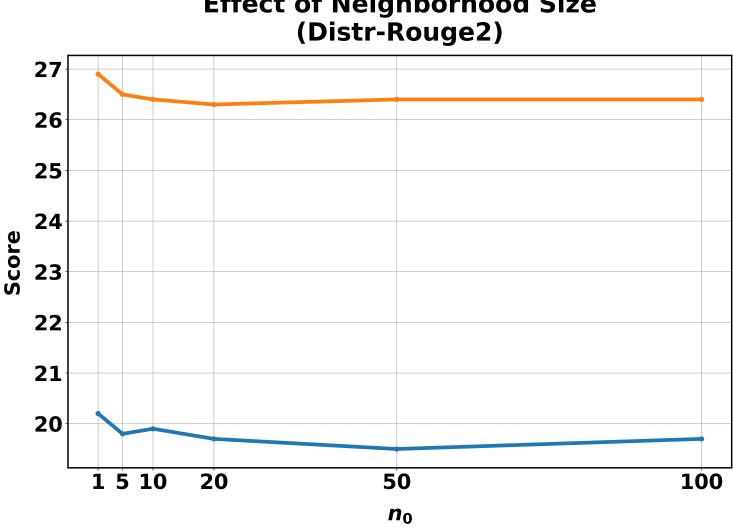

**Figure 6:** We measure how SMOOTHIE-LOCAL's performance on DISTR-ROUGE2 changes as $n_0$ changes.

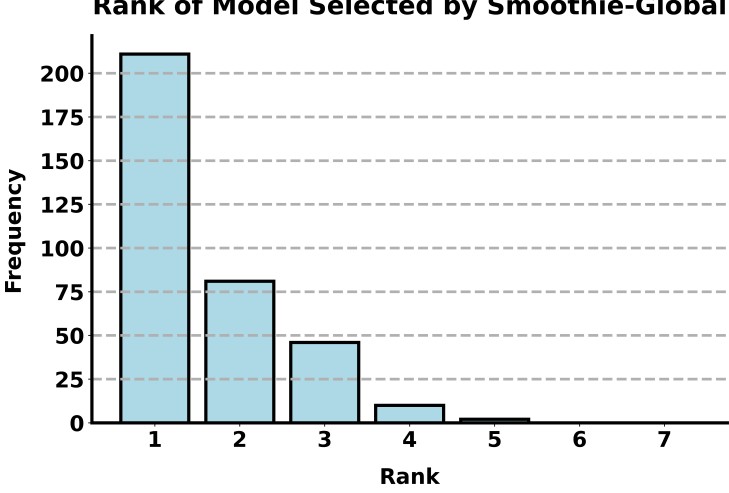

**Figure 7:** We evaluate SMOOTHIE-GLOBAL's ability to identify the best model by randomly sampling 50 ensembles of size 4-7 LLMs from a pool of the LLMs contained in the 3B and 7B ensembles. We apply SMOOTHIE-GLOBAL to select the best LLM from within each of these ensembles across the 7 NLG tasks, and measure the rank (relative to the ensemble) of the LLM selected by SMOOTHIE-GLOBAL.

| Method | ChatGPT-Rank ($\downarrow$) |
|---|---|
| RANDOM | 5.96 |
| SMOOTHIE-GLOBAL | 3.91 |

**Table 10:** Results for SMOOTHIE-GLOBAL and baselines on MixInstruct.

| Method | Accuracy |
|---|---|
| RANDOM | 28.4 |
| BEST-ON-VAL | 37.5 |
| SMOOTHIE-GLOBAL | 37.5 |

**Table 11:** Results for SMOOTHIE-GLOBAL and baselines on GSM8K. We report accuracy, with scores scaled to 0-100.

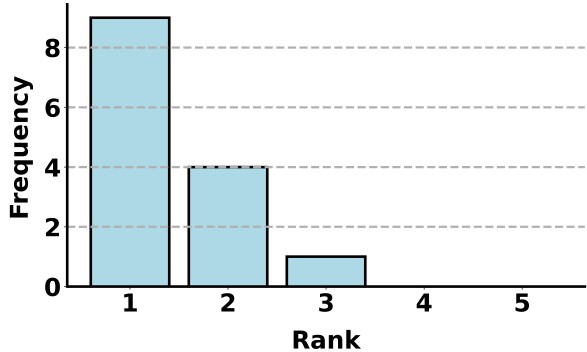

**Figure 8:** We construct a histogram over the rank of the LLM selected by SMOOTHIE-GLOBAL across both the 3B and 7B ensembles, for 7 NLG tasks.

