# OpenReview forum: "Smoothie: Label Free Language Model Routing"
_NeurIPS.cc/2024/Conference — NeurIPS 2024 poster_

### Official Review · Reviewer_gxgv · 2024-07-12

**Soundness:** 2
**Presentation:** 3
**Contribution:** 3
**Rating:** 5
**Confidence:** 4

**Summary:**

This paper proposes a label-free routing method, Smoothie, to route an ensemble of LLMs without annotated data. Smoothie constructs a latent variable graphical model over semantic embedding representations of observable LLM outputs and the unknown ground truth, estimates the sample-independent quality scores of each LLM, and routes to the LLM with the highest quality score. Experimental results indicate that Smoothie competently performs several generation tasks over the 3B and 7B ensembles.

**Strengths:**

1. Smoothie utilizes a latent graphical model and an embedding language model to estimate the quality of each separate LLM without labels, thus is more applicable than the supervised routing metods.
2. The experimental results indicate that Smoothie can outperform other supervised routing methods on several generation tasks and can also conduct prompt selection.

**Weaknesses:**

1. The method relies on the embedding method to measure the semantic similarity, which limits the scope of Smoothie to semantic-related tasks. The capability of LLMs should vary more on tasks such as mathematical reasoning, but these tasks depend less on semantic similarities.
2. The experimental comparisons are unclear. The chosen metrics are highly abstractive. Given the performances of each LLM are unknown, it's hard to identify whether the method successfully routes to the best or slightly better model than the baselines. The presentation of the experiment section is also hard to follow.

**Questions:**

1. (Refer to W1 and W2) Can you provide the performance statistics of the models used in the experiments?
2. Does Smoothie require more computation than the supervised methods?

**Limitations:**

The limitations and broader impacts are not discussed in the paper.

---

> ### Author Rebuttal · Authors · 2024-08-06
>
> We thank the reviewer for their thoughtful comments and for engaging with our work. We are grateful they appreciated (1) the generality of Smoothie’s algorithm beyond supervised methods, and (2) the breadth of experimental results.
>
> We refer the reviewer to our general response for more information on the performance statistics for individual models in the experiments (W2), as well as a discussion of the limitations. We apologize for the lack of presentation clarity in the current version of the experimental section. We hope that revisions in response to this review and others will improve clarity.
>
> Our response here addresses concerns regarding (1) the types of tasks Smoothie can be used for (W1), and (2) whether Smoothie requires more computation than supervised methods (Q2).
>
> **Performance on non-semantic tasks**
>
> The reviewer noted that because Smoothie relies on embeddings of model generations to learn quality scores for each model, Smoothie is inherently limited by the types of relationships those embeddings can capture. We agree with this observation, and have mentioned it in the limitations in the global response. However, we find that current embedding models do allow Smoothie to perform well on tasks like mathematical reasoning, as shown by our evaluation on GSM8K in the global response.
>
> **Does Smoothie require more computation than supervised methods?**
>
> Supervised methods require engineers to expend computation in two ways. First, engineers must produce generations for the ensemble over a training set. Second, engineers must train a router–typically using SGD–to map queries to models in the ensemble. At test time, supervised routers only expend computation performing inference for the selected model, i.e., one model out of the ensemble.
>
> Smoothie does not require the creation of a large training dataset, and operates solely on test-time generations without annotations. Fitting Smoothie’s weights is significantly cheaper than training a router via SGD, with our method taking only seconds in practice due to their closed form in Algorithm 1. However, Smoothie requires a generation from each model in the ensemble for the test sample, which supervised methods do not.
>
> Fortunately, the need for computing all model generations per test sample can be removed with a small algorithmic tweak, making Smoothie even more efficient and its runtime independent of $n$. Suppose we have a held-out set of $n_{train}$ train samples with precomputed generations from the models in the ensemble. For each test sample, we retrieve the most similar train samples, learn the Smoothie weights for the sample using the corresponding train sample generations, and return the model with the highest Smoothie weight (i.e., in line 5 in Algorithm 1, KNN is now over a held-out training dataset). The benefit here is that Smoothie selects the model for a test sample without needing model generations for that sample. Only $n_{train} \times m$ generations are needed, regardless of how large the test dataset $n$ is.
>
> Smoothie is still effective with this modification, which we refer to as Smoothie-Train. For instance on the 3B and 7B ensemble (using a train set of size 250, compared to test sets of size 1000), Smoothie-Train identifies the best performing model on 8/14 single-task datasets, outperforming a random selection baseline by up to 7.1 points rouge2 and 12.5 points accuracy. On the multi-task datasets across both the 3B and 7B ensemble, Smoothie-Train outperforms random sampling by an average of 3.7pts accuracy on the accuracy tasks, and by an average of 1.6pts rouge2 on the rouge2 tasks.

---

> > ### Comment · Reviewer_gxgv · 2024-08-11
> >
> > I appreciate the authors for providing (1) details about the routing, (2) performance on the non-semantic task, and (3) inference time analysis.
> > I'm particularly interested in why Smoothie can perform well on GSM8K, and whether it can perform well on reasoning datasets with questions generated from templates (similar questions with different values).
> > Still, the newly added limitations discuss the problem of semantic-based embedding.
> > My other concerns are addressed and I will raise the rating.

---

> ### Author Response · Authors · 2024-08-12
>
> Dear reviewer gxgv,
>
> Thank you so much for raising your score. We agree that the performance on GSM8K is interesting and will update our final draft with these results!

---

### Official Review · Reviewer_Cr26 · 2024-07-12

**Soundness:** 3
**Presentation:** 3
**Contribution:** 3
**Rating:** 5
**Confidence:** 3

**Summary:**

This paper proposes a method for selecting LLMs' responses for generative tasks.

It can essentially be viewed as a "truth inference" problem in the research community of "weak supervision" and "crowdsourcing"; unlike ordinary truth inference methods, this paper takes into account unstructured textual information and, for each sample, the method in this paper needs to pick one out of all candidate answers.

**Strengths:**

1) First, the problem investigated in this paper---selecting LLMs' responses for generative tasks---is very practical and interesting.

2) The proposed method is generally very intuitive and I think it will be effective.

**Weaknesses:**

Main concerns:

1) First, authors should summarize the most relevant works in the main text of the paper. For this current version, I need to scrutinize the most relevant works in the appendix.

2) Related to the above point, there are already works that focus on the task that this paper addresses, e.g. "An error consistency based approach to answer aggregation in open-ended crowdsourcing".
(Although this paper focuses on crowdsourcing workers rather than LLMs.)
 Therefore, it would be advisable for the authors to research the relevant literature in more depth and to consider them in experiments as comparison methods.

3) The core theoretical part of this paper is applied to the content in the existing work [72]. In the main text, it is necessary to state the technical and theoretical differences with the existing work [72] more clearly and in detail.

Some other concerns:
1) The meaning of some symbols is not explained, e.g., line 109.
2) In this paper, a graphical model is presented, then a graphical representation of this graphical model should be shown.
3) Typos. E.g., "x(Section 4.2)" in line 134, "$\theta_i(x)$s" in line 137.
4) The quality of the images (e.g. sharpness) can be further improved.
Also, in Figure 2, the name of the proposed method should be shown in capital letters.
5) The tables are not self-explanatory enough, e.g. the metric of interest is not shown in Table 1.
6) Some of the presentations are confusing, such as lines 609 and 610.

**Questions:**

Please refer to the "main concerns" above.

**Limitations:**

Yes.

---

> ### Author Rebuttal · Authors · 2024-08-06
>
> We thank the reviewer for their detailed feedback and are glad that they found the problem interesting and practical. Below, we address the reviewer’s concerns around related work and baselines, differences with Shin et. al., and writing clarifications.
>
> **Related work**: Thank you for your suggestion on moving more related works up into the body. We will add relevant works on routing and ensembling to the body in our updated draft.
>
> We compare and evaluate the AEC method presented in the paper mentioned in W2. In this paper, each generation has two embeddings, a global embedding (Universal Sentence Encoder) and a local one (GLEU). They solve an optimization problem that estimates each generation’s error by constructing a loss that enforces that the local and global embedding similarity between the true generation (produced by a weighted average) and the candidate generation should be the same. In contrast, Smoothie uses one embedding space, relies on a multivariate Gaussian structure among embeddings, and does not require gradient descent to learn the weight parameters.
>
> To evaluate AEC, we implemented it ourselves since we were unable to find a codebase online. We used 100 epochs and a learning rate of 0.001 for all datasets. We find that Smoothie outperforms AEC on multi-task datasets by 0.9 points on average, and on single-task datasets by 2.3 points on average across 3B and 7B ensembles. We will update our draft with the discussion and empirical results on AEC.
>
> Finally, in the spirit of the reviewer’s requests for additional baselines, we also report a comparison to PairRM. We refer to the global response for a description of this baseline’s performance.
>
> **Comparison with Shin et. al.**: Both Smoothie and Shin et. al. use a multivariate Gaussian model. However, in Smoothie we apply it to model routing with SBERT embeddings on natural language datasets, whereas Shin only conducts synthetic experiments in hyperbolic spaces and metric spaces induced by synthetic graphs. Moreover, Smoothie uses nearest neighbor kernel smoothing to allow for sample-dependent weights—critical for routing—while Shin only calculates one global set of weights over the dataset. We will add this comparison to the related work in the body of our paper.
>
>
> **Writing clarifications**: We thank you for pointing these writing errors out and apologize for them. We will address them in our updated draft.

---

> > ### Author Response · Authors · 2024-08-12
> >
> > Dear reviewer Cr26,
> >
> > We greatly appreciate your detailed comments on our work. We hope that our response has addressed your concerns about comparisons to other works and writing clarifications. In particular, we implemented the algorithm in the error consistency paper you mentioned as well as an additional reward model baseline. We compare these methods to Smoothie in our rebuttal to you and in the global rebuttal, respectively. Since the discussion period is ending in 2 days, please let us know if there are any additional questions or comments you have. Thank you so much!

---

> > > ### Comment · Reviewer_Cr26 · 2024-08-13
> > > **Response**
> > >
> > > Thanks to the authors for making the response, including adding some valuable experiments.
> > > Most of my main concerns have been addressed.
> > >
> > >
> > > Though there are still a few details: In the authors' response, there is no explanation of which method "PairRM" refers to, and there is no explanation of whether it is possible to add a graphical representation of the proposed graphical model to the text.

---

> > > > ### Author Response · Authors · 2024-08-13
> > > >
> > > > Dear reviewer Cr26,
> > > >
> > > > Thank you for your response! We are glad to hear that our additional experiments and comments addressed most of your main concerns. We apologize for those that were unaddressed.
> > > >
> > > > First, we intend to update the draft to remedy and include the suggestions you made. We will add a visual depiction of the PGM to the paper. Smoothie's PGM is similar to the "label model" in [this image](https://cdn.snorkel.ai/wp-content/uploads/2022/03/Weak-supervision-modeling-pipeline-Snorkel-Blog-1024x576.png) where the vertices in our case are $V = \{\lambda_1(x), \dots, \lambda_m(x), z^\star(x)\}$ and the edges are $E = \{(\lambda_i(x), z^\star(x))\}_{i=1}^m$. We will also update our draft with higher quality images.
> > > >
> > > > Second, we apologize for the lack of clarity regarding PairRM. PairRM is a pretrained reward model introduced and described by Jiang et al here: https://arxiv.org/abs/2306.02561. It is built from the DeBERTa model (400M parameters), and accepts  an instruction $i$, and a set of candidate responses to the instruction $r_1, ..., r_k$. PairRM then produces a ranking over these candidate responses, according to the predicted quality of the response (i.e., the best response is ranked first). We choose to evaluate PairRM because of its strong performance; it has been used to rank responses (in conjunction with KTO/DPO methods) to allow for 7B models, such as Contextual AI's model, to do well on the [AlpacaEval leaderboard](https://tatsu-lab.github.io/alpaca_eval/), where the top models are typically much larger. PairRM can be downloaded from huggingface here: https://huggingface.co/llm-blender/PairRM

---

### Official Review · Reviewer_r3vp · 2024-07-12

**Soundness:** 3
**Presentation:** 2
**Contribution:** 3
**Rating:** 6
**Confidence:** 3

**Summary:**

- This work proposes a method called SMOOTHIE, which can route label-free test examples to LLMs. Specifically,
  - it employs a latent variable model and Gaussian distribution for efficient quality score estimation and uses LLM outputs to estimate generator quality.
  - it estimates specific to each test sample using nearest neighbors and routes samples to LLMs with highest quality scores.

- Empirical results show that:
1. SMOOTHIE's learned quality weights correlated with actual LLM performance.
2. In mixed-tasks datasets SMOOTHIE is able to route different samples to different LLMs which boosts the performance.
3. SMOOTHIE can be used for prompt-selection.

**Strengths:**

- With the recent advances in LLM research, how to choose the best LLM/ how to select the best prompt for different tasks is an interesting topic and this work proposes an approach to deal with this problem.
- The empirical results show that SMOOTHIE does help increase the performance for a dataset with mixed-tasks for different embedding models and different neighborhood sizes chosen.

**Weaknesses:**

- In figure 3. there is no label on x-axis
- typo: in Section 5.2, line 263 and 264, the "SMOOTHIE-independent" be SMOOTHIEGLOBAL instead.

**Questions:**

- In the algorithm, it is not quite clear how to select indices j and k different from i. Is it just random selection?
- How do you compare SMOOTHIE with Minimum Bayes method, where you choose the output from an LLM that aligns the most with all others?

**Limitations:**

- The authors did not address the limitations
- There is no negative societal impact

---

> ### Author Rebuttal · Authors · 2024-08-06
>
> We thank the reviewer for their feedback and are happy that they found the topic interesting. In our global response, we discuss the limitations of Smoothie, which the reviewer pointed out. Below, we address the reviewer’s comments on writing clarifications as well as the Minimum Bayes method.
>
> **Writing clarifications**: We will update the draft to address the writing errors mentioned in the weaknesses section. For selecting j and k in the Smoothie algorithm, they can be randomly selected, although to reduce variance we average over all $C(m-1, 2)$ pairs of (j, k) to get $i$’s Smoothie weight. We will clarify this in our paper.
>
>
> **Minimum Bayes Risk (MBR)**: One way of minimizing Bayes risk is to select an output that has the highest similarity to all other outputs (such as [1], mentioned in [2]). That is, the model to route to for sample $x$ is $\arg \max_i \sum_j sim(g_i(x), g_j(x))$, where $sim$ is cosine similarity, for instance.
>
> We demonstrate theoretically and empirically that Smoothie-Local with k=1 (no nearest neighbor smoothing) is conceptually similar to MBR. Note that Smoothie routes to the model with the lowest value of equation 3 in the paper. If we ignore the subtraction of $\delta_{jk}(x)$, Smoothie selects the model $\arg \min_i \sum_j \delta_{ij}(x)$; for k=1, this is the generation with the lowest embedding distance from all other generations for x. Since L2 distance is inversely correlated with cosine similarity, Smoothie-Local (k=1) is hence similar to MBR. Most importantly, we found that MBR and Smoothie-Local (k=1) matched performance on both multi-task datasets and both ensembles (a 0.0003 point average difference).
>
> Therefore, Smoothie is a more general version of the MBR rule above that can almost exactly recover MBR’s behavior when k=1. Moreover, we find that Smoothie performs better than MBR for 4 out of 7 single-task datasets when using a larger k. We will update our draft to compare to MBR.
>
>
> [1] https://aclanthology.org/D18-1449/
>
> [2] https://arxiv.org/pdf/2310.01387

---

> > ### Author Response · Authors · 2024-08-12
> >
> > Dear reviewer r3vp,
> >
> > Thank you so much for your helpful feedback in your review. We hope that our response has addressed your concerns about comparison to the Minimum Bayes method, writing clarifications, and limitations. In particular, we ran extensive experiments comparing Minimum Bayes to Smoothie, with our results summarized in our rebuttal to you. Since the discussion period is ending in 2 days, please let us know if there are any additional questions or comments you have. Thank you so much!

---

> > ### Comment · Reviewer_r3vp · 2024-08-13
> >
> > Thanks for the response and clarification to the questions I had. I think it will be good to have the additional details in an updated version.

---

### Official Review · Reviewer_RLiV · 2024-07-13

**Soundness:** 3
**Presentation:** 4
**Contribution:** 3
**Rating:** 7
**Confidence:** 4

**Summary:**

This paper presents a model for routing an input to an LLM from a pool of LLMs. The aim is to estimate which LLM will produce highest quality generation without using any labeled training data in estimating the routing model. Instead, the approach relies on weak supervision to learn the parameters of a Gaussian graphical model.

The technique is evaluated by examining the model's ability to pick the oracle best LLM from the pool on various tasks; then by evaluating it's end-to-end routing performance compared to baselines (some of which have access to labeled data); and finally by using the same technique to pick amongst a set of LLM+prompt pairs for various tasks in order to effectively do prompt selection.

**Strengths:**

- The paper makes creative use of the Gaussian graphical model from Shin et al. (2022) by using a separate embedding model (SentenceBERT) to produce an embedding representation of each {input, generation} pair from an LLM. The graphical model then provides a joint distribution over those embeddings plus the latent embedding of the {input, true output}. Inference in the model is straightforward and efficient and allows for both (1) training from unlabeled data and (2) estimation of the LLM scores conditioned on the observed input. Overall, the technique provides a simple, elegant application of Shin et al. (2022)'s model and could be easily replicated. The approach is clearly described.
- The results on correlation of the graphical model quality estimates with the oracle best LLM are fairly high.
- The end-to-end routing evaluation considers a breadth of tasks including both classification and generation problems. Overall the results tend to be on par or better than strong baselines, including those that use labeled training data for model selection.
- Finally, the application to prompt selection is a nice addition demonstrating the applicability of the approach beyond the obvious routing application.
- Each experiment considers a variety of models (as well as tasks) and most experiments include multiple pools (e.g. a 3B and 7B pool) to showcase that effects across different familys of LLMs.

**Weaknesses:**

- The paper has no discussion of efficiency, but this seems like an important point to address. This has important implications for the motivation: if one has the capacity to run a pool of small models, why wouldn't one instead run a single larger model. In this way, the paper should really address the tradeoff of accurracy/ROUGE with # of parameters (or even more realistically runtime).
- The paper does very little to discuss its limitations. There are a few comments peppered throughout.
- The paper does not mention anything about which LLMs are being picked. For example, is the job of the router trivialized by certain pools of models? For example, one could imagine a setting in which the router simply identifies and then always picks the best model. The paper does no analysis of this.

**Questions:**

- How would address the efficiency concerns mentioned in the weaknesses section?
- Did you observe anything about the behavior of the routing across different tasks?

**Limitations:**

Yes

---

> ### Author Rebuttal · Authors · 2024-08-06
>
> We thank the reviewer for their thoughtful comments and for engaging with our work. We are glad to hear they appreciated our approach and the comprehensiveness of our experiments. We refer the reviewer to our general response for more information on (1) Smoothie’s routing behavior, and (2) Smoothie’s limitations. Our individual response to reviewer RLiV focuses primarily on the concerns regarding the efficiency of routers.
>
> **Why run a pool of small models as opposed to one large model?**
>
> Reviewer RLiV asked about the tradeoff between spending resources on a single large model, as opposed to multiple small models. We believe there are several reasons why multiple smaller models might be preferred to a large model.
>
> First, an ensemble of small models may exceed the performance of a large model while incurring the same computational cost. For instance, we observe that applying Smoothie to an ensemble of three 1-2B parameter models (Qwen, Gemma, and Phi-3) outperforms each of the following 7B models by up to 20pts accuracy on SQuAD: Llama-2, Storm, Snorkel, Vicuna, Nous-Capybara, and Mistral.
>
> Second, engineers increasingly access models through APIs, and API calls for larger models can often be 4-5x the cost of API calls to smaller models. On Together for instance, Llama-3 8B costs 10c per million tokens and Llama-3 70B costs 54c per million tokens. Running an ensemble of five 8B models thus costs less than a single 70B model.
>
>
> **Can Smoothie be made more efficient?**
>
> Reviewer RLiV also asked whether it was possible to incorporate more discussion of Smoothie’s efficiency. To estimate the Smoothie weights for routing, we use a simple closed-form procedure **that does not require any SGD or training**, as described in Algorithm 1. As a result, Smoothie weights on the entire dataset can be computed in seconds—for the 7B ensemble, SmoothieLocal on the multi-task datasets takes 2.14 seconds per 1000 samples, and SmoothieGlobal on the single-task datasets takes under 0.03 seconds per 1000 samples. Moreover, Smoothie **does not require any ground-truth annotations**; however, all m model generations per test sample are needed as input to the algorithm. That is, we need $n \times m$ generations for a test dataset of n samples.
>
> Fortunately, the need for computing all model generations per test sample can be removed with a small algorithm tweak, making Smoothie even more efficient and its runtime independent of $n$. Suppose we have a held-out set of $n_{train}$ train samples with precomputed generations from the models in the ensemble. For each test sample, we retrieve the most similar train samples, learn the Smoothie weights for the sample using the corresponding train sample generations, and return the model with the highest Smoothie weight (i.e., in line 5 in Algorithm 1, KNN is now over a held-out training dataset). The benefit here is that Smoothie selects the model for a test sample without needing model generations for that sample. Only $n_{train} \times m$ generations are needed, regardless of how large the test dataset $n$ is.
>
> Smoothie is still effective with this modification, which we refer to as Smoothie-Train. For instance on the 3B and 7B ensemble (using a train set of size 250, compared to test sets of size 1000), Smoothie-Train identifies the best performing model on 8/14 single-task datasets, outperforming a random selection baseline by up to 7.1 points rouge2 and 12.5 points accuracy. On the multi-task datasets across both the 3B and 7B ensemble, Smoothie-Train outperforms random sampling by an average of 3.7pts accuracy on the accuracy tasks, and by an average of 1.6pts rouge2 on the rouge2 tasks.
>
> We will update the paper to include this discussion.

---

> > ### Author Response · Authors · 2024-08-12
> >
> > Dear reviewer RLiV,
> >
> > We appreciate your valuable feedback and suggestions. We hope that our response has addressed your questions about efficiency, analysis of routing behavior, and limitations of Smoothie. Since the discussion period is ending in 2 days, please let us know if there are any additional questions or comments you have. Thank you so much!

---

> > ### Comment · Reviewer_RLiV · 2024-08-13
> > **Reply to rebuttal**
> >
> > Thanks for the thorough response! It'll be great to have these in the next version of the paper.

---

### Author Rebuttal · Authors · 2024-08-06

We thank the reviewers for their valuable feedback. We are glad that reviewers found the Smoothie algorithm to be elegant (RLiV, Cr26, gxgv), recognized the practical applications of this work (r3vp, Cr26), and appreciated the breadth of our evaluation (RLiV).

Our global response (1) discusses results on new datasets and baselines showcasing Smoothie’s performance, (2) provides more details on Smoothie’s routing behavior, and (3) describes Smoothie’s limitations.

## Results on new datasets and baselines

__Comparison to PairRM__: We benchmark Smoothie against PairRM [1], a popular pre-trained reward model used in prior work for both generation selection and routing. Given one or more responses to an instruction, the PairRM scores are used to rank the responses by relative quality.

On the multi-task accuracy dataset, Smoothie-Local outperforms PairRM for both the 3B and 7B ensemble, by an average of 1.6pts accuracy. On the multi-task rouge2 dataset, Smoothie-Local slightly outperforms PairRM by an average of 0.2pts rouge2. That Smoothie-Local can match and even outperform PairRM is notable because PairRM is a supervised method, since it requires annotated pairwise data to train. In contrast, Smoothie requires no annotations.

__Performance on GSM8K__: Reviewer gxgv asked whether Smoothie could be applied to mathematical reasoning tasks. We run Smoothie-Global on GSM8K, a benchmark consisting of grade-school level mathematical word problems. In order to apply Smoothie-Global, we prompt models to produce a chain-of-thought style generation culminating in the final numeric answer. We evaluate Smoothie-Global on the 7B ensemble, and find it  successfully identifies the best performing model in this ensemble, producing an accuracy/solve-rate of 37.5%. In contrast, a random-selection baseline over the ensemble produces a solve-rate of 20.1%, and using PairRM to select from within the ensemble produces a solve-rate of 27%.


__Performance on MixInstruct__: We evaluate Smoothie-Global on MixInstruct [1], a dataset consisting of instructions and corresponding responses from 12 models, along with relative rankings of response quality. Smoothie-Global successfully identifies the best average model from the 12 ensemble models.


We will update our paper to include these results.

## Routing behavior

Reviewers RLiV and gxgv asked for more details regarding Smoothie’s routing behavior. Specific details requested include:
- The performance of individual models.
- How often Smoothie picks different models across samples.
- The relative quality of the model selected by Smoothie for each sample.

We will update our draft to include the individual performance of each model for all datasets (single task and multi-task). We briefly summarize the important findings below, and include corresponding visualizations in the attached PDF.

**(1) Smoothie-Local picks different models across multitask datasets**

For both multi-task datasets for both the 3B and 7B model groups, we observe that Smoothie-Local selects every model in the ensemble for at least one sample. The least selected model was selected 8.7% of the time, and the most selected model was selected 43% of the time. More information is in Figure 1 of the PDF.

**(2) Smoothie-Local improves upon the best ensemble model**

Smoothie-Local’s selection of different models improves performance. Smoothie-Local outperformed the best model in the ensemble for both multi-task datasets across both the 3B and 7B groups, by as much as 7pts accuracy and 1.6pts rouge2.

**(3) Per-sample, Smoothie-Local frequently selects the best model available**

For each of the multi-task datasets, we study whether the model generation selected by Smoothie-Local was the best generation that Smoothie-Local could have selected. In Figure 2 of the PDF, we provide the distribution of the per-sample rank of the generation that Smoothie selects. On the rouge2 dataset, we find that Smoothie selected the best generation possible for 36% of samples for the 3B ensemble, and for 27% of samples for the 7B ensemble. Smoothie’s selected generation was better than the median for 72% of samples for the 3B ensemble and for 79% of samples for the 7B ensemble. On the accuracy multi-task dataset, Smoothie selected the best possible generation for 85% of samples for the 3B ensemble, and 89% of samples for the 7B ensemble. Smoothie’s selected generation was better than the median for 99% of samples for both the 3B and 7B ensemble.

In short: we find that Smoothie selects a diverse array of models, and that selection of multiple models improves performance.


## Limitations

Reviewers RLiV, r3vp, and gxgv noted that our submission did not mention Smoothie’s limitations. We apologize for this omission. We discuss limitations of Smoothie below and include them in the Discussion section of our updated draft.

1. The multivariate Gaussian model uses a diagonal covariance matrix. Smoothie thus assumes that the error vector for each generation is independent, although Smoothie can be extended to learn and account for dependencies [2, 3].
2. Smoothie only optimizes for performance, not cost. Recent works focus on the cost-performance tradeoff when routing among large costly models and small cheaper models. We consider cost optimization as future work.
3. Smoothie relies on embeddings, which may only capture particular aspects of semantic similarity among generations, to determine quality. Other embedding models and additional heuristics could be used to create richer input features for Smoothie.

[1] https://arxiv.org/abs/2306.02561

[2] https://arxiv.org/abs/1810.02840

[3] https://arxiv.org/abs/1903.05844

---

### Decision · Program_Chairs · 2024-09-25

**Decision:**

Accept (poster)

**Comment:**

This paper introduces a method for routing inputs to the most suitable LLM from a pool of models. The goal is to predict which LLM will produce the highest-quality output without relying on labeled training data for the routing mechanism. Instead, the approach utilizes weak supervision to learn the parameters of a Gaussian graphical model with latent variables.

The method is interesting, and the results appear promising. During the rebuttal, the authors adequately addressed most of the concerns raised by the reviewers. Therefore, I vote for acceptance.